# A Mass Spectrometry-Based Proteome Study of Twin Pairs Discordant for Incident Acute Myocardial Infarction within Three Years after Blood Sampling Suggests Novel Biomarkers

**DOI:** 10.3390/ijms25052638

**Published:** 2024-02-24

**Authors:** Hans Christian Beck, Asmus Cosmos Skovgaard, Afsaneh Mohammadnejad, Nicolai Bjødstrup Palstrøm, Palle Fruekilde Nielsen, Jonas Mengel-From, Jacob Hjelmborg, Lars Melholt Rasmussen, Mette Soerensen

**Affiliations:** 1Center for Individualized Medicine in Arterial Diseases, Department of Clinical Biochemistry, Odense University Hospital, J. B. Winsloews Vej 4, 5000 Odense, Denmark; hans.christian.beck@rsyd.dk (H.C.B.); nicolai.bjodstrup.palstrom@rsyd.dk (N.B.P.); palle.fruekilde@rsyd.dk (P.F.N.); lars.melholt.rasmussen@rsyd.dk (L.M.R.); 2The Danish Twin Registry and Epidemiology, Biostatistics and Biodemography, Department of Public Health, University of Southern Denmark, Campusvej 55, 5230 Odense, Denmark; acskovgaard@health.sdu.dk (A.C.S.); amohammadnejad@health.sdu.dk (A.M.); jmengel-from@health.sdu.dk (J.M.-F.); jhjelmborg@health.sdu.dk (J.H.); 3Department of Clinical Genetics, Odense University Hospital, J. B. Winsloews Vej 4, 5000 Odense, Denmark

**Keywords:** acute myocardial infarction, tandem mass spectrometry, proteome-wide association analysis, discordant twin pairs

## Abstract

Acute myocardial infarction (AMI) is a major cause of mortality and morbidity worldwide, yet biomarkers for AMI in the short- or medium-term are lacking. We apply the discordant twin pair design, reducing genetic and environmental confounding, by linking nationwide registry data on AMI diagnoses to a survey of 12,349 twins, thereby identifying 39 twin pairs (48–79 years) discordant for their first-ever AMI within three years after blood sampling. Mass spectrometry of blood plasma identified 715 proteins. Among 363 proteins with a call rate > 50%, imputation and stratified Cox regression analysis revealed seven significant proteins (FDR < 0.05): FGD6, MCAM, and PIK3CB reflected an increased level in AMI twins relative to their non-AMI co-twins (HR > 1), while LBP, IGHV3-15, C1RL, and APOC4 reflected a decreased level in AMI twins relative to their non-AMI co-twins (HR < 1). Additional 50 proteins were nominally significant (*p* < 0.05), and bioinformatics analyses of all 57 proteins revealed biology within hemostasis, coagulation cascades, the immune system, and the extracellular matrix. A protein–protein-interaction network revealed Fibronectin 1 as a central hub. Finally, technical validation confirmed MCAM, LBP, C1RL, and APOC3. We put forward novel biomarkers for incident AMI, a part of the proteome field where markers are surprisingly rare and where additional studies are highly needed.

## 1. Introduction

Cardiovascular disease (CVD) is the leading cause of mortality in humans; nearly 20 million individuals worldwide die from cardiovascular events every year [1,2]. Acute myocardial infarction (AMI) occurs almost exclusively due to atherosclerotic occlusion of a coronary artery, cutting off blood supply and leading to damage to the heart muscle [3]. The underlying arterial pathology, atherosclerosis, develops over decades, including the initial buildup of fatty streaks, which develop into inflammatory fibrofatty lesions and vulnerable plaques as the basis for acute vascular occlusions [4]. Despite considerable improvements in prevention, diagnosis, and treatment strategies for AMI over the last many decades, AMI remains one of the major causes of mortality and morbidity worldwide; the prevalence of AMI approaches three million people worldwide and more than one million deaths each year in the United States alone [3].

A priori hypothesis-driven studies of blood plasma molecules have led to several useful biomarkers of AMI. These biomarkers serve two different purposes: either they are used for the diagnosis of a present AMI or they are used to foresee the future risk of AMI. For the biochemical diagnosis of present AMI, the increased levels of the cardiac muscle proteins troponin T and troponin I and the creatinine kinase (CK) isoforms CK1/CK-BB and CK2/CK-MB, reflecting damage to the myocardium, are well defined and widely used in the daily clinic [5]. The search for protein biomarkers that can predict future AMI has, on the other hand, been less successful [6]. The conventional risk factors for cardiovascular events, first of all lipid biomarkers (e.g., low density lipoprotein (LDL) and non-high density lipoprotein (non-HDL)), inflammatory biomarkers (e.g., CRP), and certain genetic variants [6,7] have been known for decades and have been built into risk scores like the Framingham Heart Score, where they display a moderate efficiency in pointing out individuals at future risk on a long-term basis, but they are not very efficient regarding the short- or medium-term basis [8,9]. Therefore, the discovery of protein biomarkers for earlier-occurring AMI is warranted, and such biomarkers are potentially useful in a clinical setting, for example, in relation to indications for imaging and invasive coronary diagnostics.

Contrary to a priori hypothesis-driven candidate protein studies, some studies have applied a hypothesis-free approach, exploring all proteins present in blood plasma. While initial studies applied laboratory techniques such as two-dimensional gel electrophoresis followed by mass spectrometry (MS), e.g., [10,11], recent improvements in MS methodology have enabled more unbiased and comprehensive analyses [12]. To the best of our knowledge, only the following three MS-based studies have investigated blood plasma from AMI patients and controls: Xu et al. (2019) [2] measured 468 proteins in 10 patients and 5 controls; Pan et al. (2020) [13] measured 950 proteins in 24 patients and 8 controls; and Xie et al. (2022) [14] measured 789 proteins in 8 patients and 8 controls. In total, these studies reported 33, 95, and 72 differentially expressed proteins, respectively. As seen from these studies, AMI proteome studies have been case-control studies; consequently, these studies investigate blood samples drawn after the AMI has occurred and do therefore not enable the study of incident cases and the identification of biomarkers predicting future AMI events. Hence, more prospective studies of AMI, also considering issues such as multiple testing, are needed.

Finally, a challenge in attempts to identify relevant plasma biomarkers for AMI is the difficulty of separating such biomarkers from the well-known risk factors of AMI, for example, blood lipid levels, and from the influence of relevant genetic factors. Genetic studies of AMI have reported several relevant genes [15], as have genetic studies of several of the phenotypes considered AMI risk factors, e.g., blood lipid levels, hypertension, smoking habits, obesity, and diabetes mellitus. Moreover, several studies have shown in recent years that the concentrations of a substantial part of the plasma proteins are strongly influenced by genetic determinants [16,17]. Consequently, the association of protein biomarkers with AMI outcomes might be confounded by genetic variation. One way to reduce such genetic bias is to investigate twin pairs, where one twin develops the disease of interest while the co-twin does not; this study design is known as the discordant twin pair design [18]. This study design is statistically very powerful; for instance, it has been estimated that for epigenome-wide association studies of more than 450,000 DNA methylation sites, only 98 monozygotic twin pairs are needed for detecting a 10% methylation difference with a power of 80% [19], and that the classical case-control study needs a minimum of 10 times as many individuals as investigating monozygotic twin pairs [20]. In addition to genetics, differences in environmental factors potentially confounding the association of protein biomarkers with AMI outcomes are also controlled in the discordant twin-pair design, especially factors related to the shared early-life environment.

The aim of the current study is to identify proteins associated with medium-term (<3 years) incident cases of AMI by applying the discordant twin pair design in the population-based INFRA twin survey of middle-aged Danish twins of the Danish Twin Registry (DTR) [21]. By linking the 12,349 twins of the INFRA cohort to the nationwide National Danish Patients Registry, 39 discordant twin pairs were identified, of which 12 twin pairs were monozygotic, 25 twin pairs were dizygotic, and 2 twin pairs were of unknown zygosity. Blood plasma samples from these twins were investigated by nano-liquid chromatography coupled to tandem mass spectrometry (nano-LC-MS-MS). To the best of our knowledge, the present study is the first of its kind for the identification of protein biomarkers of incident AMI in twins.

## 2. Results

Seventy-eight twins belonging to 39 twin pairs were investigated in the present study. These twin pairs had been identified via the nationwide Danish Patient Registry as discordant for their first-ever AMI diagnosis within the first three years after blood sampling. Blood plasma samples from these twins were analyzed by mass spectrometry (MS) for proteome analysis and analysis of cotinine levels, as well as by standard biochemical analysis for lipid levels.

### 2.1. Time to Acute Myocardial Infarction Diagnosis and Protein Levels in the 39 Twin Pairs

Descriptives of the study population can be seen in Table 1. The 39 co-twins receiving an AMI diagnosis within the first three years after blood sampling did so with an average time from blood sampling to diagnosis of 1.39 years (SD = 0.83, range: 0.01–2.9 years, median = 1.49 years). None of the 39 AMI-free co-twins died within the three years of follow-up, while three of the AMI co-twins died. In total, 715 proteins were measured in the MS analysis. Of these proteins, 197 were measured in all individuals (i.e., displayed a call rate of 100%), while for the remaining 518 proteins, 63 proteins had a call rate of >75% to <100%, 103 proteins had a call rate of >50% to <75%, 154 proteins had a call rate of >25% to <50%, and 198 proteins had a call rate of >0% to <25%. For the 197 proteins measured in all individuals, the mean protein values ranged from 0.91 (SD = 0.27) to 1.35 (SD = 1.04). Calculating the intra-twin pair difference in protein values (subtracting the protein value for the co-twin without AMI from the protein value from the co-twin with AMI), the mean differences in protein values ranged from −0.24 (SD = 1.06) to 0.18 (SD = 0.94). Finally, when inspecting the intra-twin pair correlations of the 197 proteins measured in all individuals, 128, 49, and 46 proteins were found to be significantly (*p* < 0.05) more correlated within the twin pairs than between the twin pairs for the monozygotic (N = 12), the dizygotic same gender (N = 12), and the dizygotic opposite gender twin pairs (N = 13), respectively, indicating a genetic influence on protein levels. On average, the intra-twin pair correlations were 0.67 (SD = 0.14, range: 0.46–0.95), 0.63 (SD = 0.12, range: 0.45–0.92), and 0.56 (SD = 0.09, range: 0.44–0.77) for the monozygotic, the dizygotic same gender, and the dizygotic opposite gender twin pairs, respectively.

### 2.2. Association Analysis of Time-To-Diagnosis and the Proteome Data in the 39 Twin Pairs

As the present study population was composed of AMI discordant twins, it is possible to reduce the potential confounding induced by shared environmental factors and genetics by investigating the intra-twin pair differences in protein levels. Analysis of the 363 proteins with a call rate above 50% in the 39 twin pairs revealed 57 proteins with a nominally significant *p* value (*p* value < 0.05), and of these, seven proteins passed correction for multiple testing (FDR *p* value < 0.05) (see Table 2 and Appendix A). Among these seven proteins, the FYVE, RhoGEF, and PH domain-containing protein 6 (FGD6), the cell surface glycoprotein MUC18 (MCAM), and the phosphatidylinositol 4,5-bisphosphate 3-kinase catalytic subunit beta isoform (PIK3CB) revealed a hazard ratio (HR) above 1 reflecting increased level of the protein in the AMI twins relative to the non-AMI co-twins, while the lipopolysaccharide-binding protein (LBP), the immunoglobulin heavy variable 3-15 (IGHV3-15), the complement C1r subcomponent-like protein (C1RL) and the apolipoprotein C-IV (APOC4) displayed a HR below 1 reflecting decreased level of the protein in the AMI twins relative to the non-AMI twins. Of the 50 proteins with a *p* value below 0.05, yet not passing correction for multiple testing, 19 displayed a HR < 1, while 31 showed a HR > 1 (see Appendix A).

Bioinformatic analyses of the seven proteins passing correction for multiple testing did not reveal any significant findings. When including all 57 proteins, 5 immunoglobulins were not annotated in the STRING database; consequently, pathway analyses were performed for the remaining 52 proteins. These analyses revealed 5 KEGG and 11 Reactome pathways (see Table 3), as well as 12 STRING clusters, 5 GO Functions, 22 GO Components, and 23 GO Processes (see Appendix A). The KEGG and Reactome pathways reflect, among others, hemostasis, the immune system, and extracellular matrix (ECM) biology (see Table 3). This was also reflected in the 12 STRING clusters, which all rooted back to three clusters, namely ‘Mixed, incl. Complement and coagulation cascades, and Serine-type endopeptidase activity’, ‘Mixed, incl. Immunoglobulin complex, and Immunoglobulin binding’ and ‘Extracellular matrix organization, and Biomineralization’ (see Appendix A). Finally, the 23 GO Processes foremost reflected immune biology, inflammation, and cellular processes, such as cell migration, while the 5 GO Functions reflected binding activity, and most of the 22 GO Components reflected cellular anatomical entities related to the cell surface or extracellular spaces or transport, as well as protein-containing complexes.

The protein–protein interaction network (PPI) based on the 52 proteins with a *p* value below 0.05 is shown in Figure 1. In this network, all but 10 proteins had connections to other proteins, and Fibronectin 1 (FN1) was the most connected protein. The PPI reflected three overall sub-clusters as follows: one sub-cluster including several immunoglobulins (bottom of figure), one large sub-cluster with FN1 in the center (top of figure), and one smaller sub-cluster including, among others, APOC4 and C1RL (right side of figure).

Notes: (a) 42 proteins have at least one connection (displayed in figure), while (b) 10 do not have connections), and (c) 5 proteins are not annotated in the STRING database).(a)ACTN1: alpha-actinin-1; ANK3: ankyrin-3; ANPEP: aminopeptidase N; APOC4: apolipoprotein C-IV; ART4: ecto-ADP-ribosyltransferase 4; C1RL: complement C1r subcomponent-like protein; CD5L: antigen-like CD5; CFL1: cofilin-1; CLTCL1: clathrin heavy chain 2; COL6A3: collagen alpha-3(VI) chain; FCGR3A: low-affinity immunoglobulin gamma Fc region receptor III-A; FN1: fibronectin; GPD1: glycerol-3-phosphate dehydrogenase [NAD(+)];GPLD1: phosphatidylinositol-glycan-specific phospholipase D; HABP2: hyaluronan-binding protein 2; HSPA5: endoplasmic reticulum chaperone BiP; IGHV3-15: immunoglobulin heavy variable 3-15; IGHV3-72: immunoglobulin heavy variable 3-72; IGKV1D-33: immunoglobulin kappa variable 1D-33; IGLL1: immunoglobulin lambda-like polypeptide 1; INPP4B: inositol polyphosphate 4-phosphatase type II; JCHAIN: immunoglobulin J chain; LCAT: phosphatidylcholine-sterol acyltransferase; LYZ: lysozyme C; MCAM: cell surface glycoprotein MUC18; MSN: moesin; NRCAM: neuronal cell adhesion molecule; PCOLCE: procollagen C-endopeptidase enhancer 1; PCYOX1: prenylcysteine oxidase 1; PIGR: polymeric immunoglobulin receptor; PIK3CB: phosphatidylinositol 4,5-bisphosphate 3-kinase catalytic subunit beta isoform; PRX: periaxin; PTPN1: tyrosine-protein phosphatase non-receptor type 1; QSOX1: sulfhydryl oxidase 1; SELL: L-selectin; SERPINA10: protein Z-dependent protease inhibitor; SERPINF2: alpha-2-antiplasmin; STAT1: signal transducer and activator of transcription 1-alpha/beta; THBS4: thrombospondin-4; TLN1: talin-1; TPI1: isoform 2 of triosephosphate isomerase; VTN: vitronectin.(b)CFD: complement factor D; DDX19B: ATP-dependent RNA helicase DDX19B; FGD6: FYVE, RhoGEF, and PH domain-containing protein 6; GLIPR2: Golgi-associated plant pathogenesis-related protein 1; LBP: lipopolysaccharide-binding protein; LRRC17: leucine-rich repeat-containing protein 17; PCDH15: isoform 4 of Protocadherin-15; QTRT2: queuine tRNA-ribosyltransferase accessory subunit 2; SYMPK: symplekin; ZNF268: zinc finger protein 268.(c)IGLV1–40: immunoglobulin lambda variable 1–40; IGLV1–47: immunoglobulin lambda variable 1–47; IGLV2–23: immunoglobulin lambda variable 2–23; IGLV3–16: immunoglobulin lambda variable 3–16; IGLV6–57: immunoglobulin lambda variable 6–57.

### 2.3. Investigation of the Direction of Effect of the 57 Nominal Significant Proteins in the 12 Monozygotic Twin Pairs

Of the 39 twin pairs, 12 were monozygotic, hence enabling the study of intra-twin pair differences in protein levels in genetically identical individuals, an analysis that excludes the potential for genetic confounding. This means that such an analysis can potentially confirm the direction of the effect of proteins not affected by genetic variation. The 57 proteins found to be nominally significant in the analysis of all twin pairs were examined in the monozygotic twins; 3 proteins could not be analyzed due to missing data, yet of the 54 remaining proteins, 47 reflected the same direction of effect in the monozygotic twins as in all twins, while 7 proteins reflected the opposite direction of effect. Of the 47 proteins with the same direction of effect, 14 proteins displayed a *p* value below 0.05 in the analysis of the monozygotic twins, while of the 7 proteins with the opposite direction of effect, 2 proteins displayed a *p* value below 0.05 in the analysis of the monozygotic twins (see Appendix A). The PPI of the 47 proteins, which showed the same direction of effect in the monozygotic twins as in all twins, reflected the same overall pattern as seen in all twin pairs (see Appendix A). Lastly, when including the 47 proteins in bioinformatic analyses (see Appendix A), many of the same biological pathways and processes were seen; of the 5 KEGG pathways, 11 Reactome pathways, 12 STRING clusters, 5 GO Functions, 22 GO Components, and 23 GO Processes observed in all twin pairs, 5 (100%), 9 (82%), 11 (92%), 5 (100%), and 18 (82%), respectively, 9 (39%) were reflected also in the monozygotic twins. Many of the GO Processes not reflected in the monozygotic twin pairs related to the immune system. Taken together the monozygotic twins overall echo the results observed in all twins. 

### 2.4. Investigation of the Seven Proteins Passing Correction for Multiple Testing by Parallel Reaction Monitoring Mass Spectrometry

Finally, for experimental validation of the seven proteins passing correction for multiple testing (i.e., the proteins in Table 2), we inspected these proteins by parallel reaction monitoring mass spectrometry (PRM-MS). As a positive control, we also investigated the APOA1 and APOB proteins, often investigated in relation to CVD and known to correlate to HDL and LDL values, respectively. As seen in Appendix A, both APOA1 and APOB displayed good statistically significant correspondence between nano-LC-MS-MS and PRM-MS, with correlation coefficients of 0.83 (APOA1) and 0.86 (APOB), respectively. Furthermore, both types of MS data also showed statistically significant correlations to HDL and LDL values, with correlation coefficients of 0.78 (nano-LC-MS-MS) and 0.84 (PRM-MS) for APOA1 and HDL and 0.87 (nano-LC-MS-MS) and 0.78 (PRM-MS) for APOB and LDL. When investigating the seven proteins passing correction for multiple testing, four of the following proteins displayed good correspondence between the two MS methods: correlation coefficients: MCAM = 0.44, LBP = 0.62, C1RL = 0.35, and APOC4 = 0.79 (see Appendix A). The remaining three proteins, i.e., IGHV3-15, FDG6, and PIK3CB, could not be measured by PRM-MS.

## 3. Discussion

Acute myocardial infarction (AMI) remains one of the major causes of mortality and morbidity worldwide, and biological markers for identification of short- and medium-term incident AMI are warranted given that the present markers are not very precise and are difficult to apply on the individual level [8,22]. In the present study, we take advantage of applying the discordant twin pair design for this need, as this design reduces the potential confounding induced by genetics and the early environment, as well as shared environmental factors. The AMI diagnoses of the present study population were drawn from the nationwide Danish Patient Registry (DPR), hence enabling analysis of incident cases without relying on self-report.

Analysis of 39 twin pairs discordant for their first-ever AMI within three years after blood sampling revealed 7 proteins passing correction for multiple testing and an additional 50 proteins with nominal significance. Bioinformatic analyses revealed several biological pathways, foremost related to hemostasis, the coagulation cascade, the immune system, and extracellular matrix (ECM) biology. Of the seven proteins passing correction for multiple testing, three have previously been put forward as cardiovascular makers: apolipoprotein C-IV (APOC4) is a lipid-binding protein belonging to the apolipoprotein gene family, and the gene is located in the so-called *APOE*-*APOC1*-*APOC4*-*APOC2* gene cluster, which has been intensively studied in relation to cardiovascular traits and risk factors, e.g., [23]. With respect to proteome studies, APOC4 has been reported to be associated with coronary atherosclerosis [24], heart failure after AMI [25], and recovery from stroke [26]. Additionally, lipopolysaccharide-binding protein (LBP) has been reported as a protein marker of AMI [13] and to be associated with coronary atherosclerotic plaque disruption [27], as well as a risk of AMI after human immunodeficiency virus [28] or COVID-19 infection [29]. LBP plays a role in the innate immune response and belongs to a family of structurally and functionally related proteins, also holding the cholesteryl ester transfer protein and the phospholipid transfer protein. Lastly, the cell surface glycoprotein MUC18 (MCAM), also known as CD146, is a cell adhesion molecule that, among others, plays a role in wound healing as well as in the cohesion of endothelial cells in the intercellular junctions in vascular tissue. MCAM was first reported for melanoma cancer, yet also as a marker for circulating endothelial cells relevant for several CVD traits, including AMI [30], pulmonary congestion after acute coronary syndrome [31], and ischemic rest pain after peripheral atherosclerosis [32]. Furthermore, MCAM is characterized as a cardiovascular disease protein in studies by the Framingham Heart Study, e.g., [33]. The four remaining proteins are, to the best of our knowledge, less known in relation to CVD; however, some evidence points to their relevance. FYVE, RhoGEF, and PH domain-containing protein 6 (FGD6) is a member of the Ras-like family of Rho- and Rac proteins, which may play a role in, among others, the regulation of the actin cytoskeleton and cell shape. FGD6 is known from the Aarskog–Scott syndrome, characterized by short stature, facial abnormalities, skeletal and genital anomalies, and, in some cases, heart defects and a cleft lip. Expression quantitative trait loci in FGD6 have been reported to be associated with coronary heart disease [34] and to have an effect on CVD, potentially also reflected in variation in human lifespan [35]. The phosphatidylinositol 4,5-bisphosphate 3-kinase catalytic subunit beta isoform (PIK3CB) is an isoform of the catalytic subunit of phosphoinositide 3-kinase beta, which participates in several signaling pathways related to immune processes at the site of injury or infection. PIK3CB has been reported to be associated with recurrent cardiovascular events [36]. The immunoglobulin heavy variable 3-15 (IGHV3-15) and complement C1r subcomponent-like protein (C1RL), also related to immune response, have, to the best of our knowledge, not been linked to CVD previously. In general, little is known about the biological functions of these two proteins. Lastly, APOC4 and C1Rl were located together in one of the sub-clusters of the protein–protein-interaction network (PPI) generated in the present study, yet the most connected and very central protein was Fibronectin 1 (FN1). FN1 is a glycoprotein that is present in plasma in a dimeric form and at the cell surface and ECM in a dimeric or multimeric form and participates in several processes, including cell adhesion and migration processes in connection to embryogenesis, wound healing, blood coagulation, host defense, and metastasis. FN1 has been suggested in connection with AMI [37,38,39], as well as T-segment elevation myocardial infarction [40,41].

The fact that the pathway analyses of the present study foremost reflected immune response, homeostasis, blood coagulation, and ECM biology might not be surprising given that these processes are well-known in connection to AMI; homeostasis and blood coagulation potentially reflect blood clot formation, while inflammation and immune response are central for both the atherosclerosis preceding AMI and the activation of inflammatory responses after AMI. Finally, ECM remodeling is key for the reparative responses after AMI [4,42]. Interestingly, two of the previously published proteome studies of AMI also identified some of these processes: Xu et al. [2] identified complement and coagulation cascades investigating 10 patients and 5 controls, while Xie et al. identified processes related to response to wounding and wound healing investigating 8 patients and 8 controls [14]. As previously mentioned, these two studies were case-control studies performed after AMI had occurred, while the present study holds incident cases. Hence, it might appear somehow contradictory that biology reflecting present AMI is identified in the present study. On the other hand, it might simply reflect an ongoing atherosclerotic phenotype of the cases present at the time of blood sampling. Finally, these findings observed in all twin pairs were overall supported when restricting to the monozygotic twins of the study population, i.e., eliminating the genetic influence, although the *p* values were larger, likely due to the smaller sample size. Hence, future studies in additional twin cohorts of monozygotic twins would be advantageous, preferably including more monozygotic twins than in the present study population.

The present study has several strengths. First of all, the use of register data from a national registry enables information on incident AMI, contrary to the previous published case-control studies of blood drawn after the AMI has occurred. Furthermore, as information on diagnosis is not self-reported, it is likely less biased. Secondly, the use of twin data reduces potential confounding; to the best of our knowledge, the present study is the first to apply the discordant twin pair design in relation to a proteome study of incident AMI. Thirdly, studies of incident AMI cases are, in general, rare, calling for such studies. However, there are also limitations to the present study; the proteome analysis performed applied blood plasma samples; hence, the proteome profiles obtained reflect the overall general biology of the individuals and not protein variation specific to, for instance, heart muscle cells. On the other hand, the identification of useful protein biomarkers applicable in blood will have great potential due to the non-invasive nature of the sampling. Secondly, the present study is explorative in nature; we explored all detectable proteins by MS with a call rate above 50%, but we cannot exclude false positive findings, especially among the nominally significant observations. Thirdly, the 39 twin pairs of the present study population included 12 monozygotic twin pairs, 25 dizygotic twin pairs, and 2 twin pairs of unknown zygosity, which means that the association analysis performed on all twins does not completely remove the potential genetic confounding. The influence of the shared environment is, however, still being corrected. The findings in all twins were, however, echoed in the monozygotic twin pairs of the present study population. Nevertheless, validation in additional cohorts of monozygotic twins is warranted. Finally, despite the fact that the AMI cases in the present study were defined as not having an AMI diagnosis before blood sampling, we cannot exclude that pathological changes were already present at blood sampling, i.e., AMI develops over years, and hence even to condition on no diagnosis before blood sampling cannot make us sure that the proteome measured does not to some degree reflect initial disease.

In conclusion, we here put forward novel protein biomarkers, as well as biological pathways, for incident AMI within three years after blood sampling. The main findings were echoed when the analysis was restricted to the monozygotic twin pairs of the study population to control the genetic effects. These markers could potentially benefit in the detection of patients at risk of suffering from an AMI within a short period, yet findings should be verified in additional study populations.

## 4. Materials and Methods

### 4.1. The Study Population

The study population was retrieved from the Danish Twin Registry (DTR) Infrastructure Study (hereon called the Infra Cohort) conducted from 2008 to 2011 [21]. The Infra Cohort was a nationwide study of Danish twins selected from the birth cohorts of 1931–1969 with the overall aim to study aspects of midlife health, lifestyle, and functioning, as well as the contribution to differences in late-life health and mortality. In total, 12,676 twins took part in the Infra Cohort; of these 10,276 individuals (from the birth cohorts 1931–1969) had not previously taken part in a twin survey at the DTR, while 2400 twins (from the birth cohorts 1931–1952) had taken part in The Middle Age Danish Twins Study in 1998 and hence constituted a follow-up study within the Infra Cohort [21]. All participants received a mailed questionnaire, which they filled out in advance and handed in at an in-person meeting at one of the five survey centers located across Denmark. The questionnaire included questions regarding social status, family relations, childhood, self-rated health, self-reported diseases, mental wellbeing, lifestyle (including tobacco use, alcohol use, and exercise habits), medication use, and intellectual, cultural, and social activities. At the survey center, a comprehensive examination took place regarding health and aging, including objective measurements such as cognitive functioning, blood pressure, hand grip strength, chair stand, body height, weight, and lung function. In addition, whole blood samples were collected; in total, 12,391 of the twins donated blood samples, leaving EDTA blood plasma samples for 12,349 individuals as the basis for the present study. The whole blood samples were collected at the survey centers and shipped to the laboratory, where samples were centrifuged and divided into fractions before storage at −80 °C. This makes the INFRA cohort the largest population-based survey of twins holding blood samples conducted by the DTR to date [21].

For the present study, the inclusion criteria were: (A) twin pairs for whom both co-twins in the twin pair had taken part in the Infra Cohort and where blood plasma samples were available (N = 7566), and (B) twin pairs who were discordant for the first-ever AMI diagnosis-based diagnoses in the nationwide Danish Patient Registry (DPR) (see https://sundhedsdatastyrelsen.dk for details) within the first three years after blood sampling, i.e., one co-twin in each pair had at least one AMI diagnosis within three years after blood sampling, while their co-twin remained AMI diagnosis-free. We applied a time cut-off of three years as our overall aim of the present study was to identify biomarkers for short- or medium-term AMI, as biomarkers for such cases are lacking in the proteome field. The DPR contains all hospital discharges and outpatient visits from all Danish hospitals since 1977, with International Classification of Disease (ICD) 10 codes applied from 1994 and onwards and ICD-8 back to 1977. ICD-9 codes were never applied in the Danish health registers. AMI was defined as the ICD 10 code group DI21 (acute myocardial infarction: DI210-DI219) and the ICD8 codes 41009/41099, the latter based on the historical definition from Statistics Denmark (https://www.dst.dk (accessed on 10 November 2022)). Due to general data protection regulations (Danish and EU legal regulations) at the time of conducting this study, data from the DPR was available from the start of the DPR and until 2014, hence not allowing the investigation of later-occurring AMI. Only twin pairs where the twin with an AMI diagnosis after blood sampling was free of diagnosis before blood sampling (register data back to 1977) were included, i.e., only incident cases were included. Out of 7566 individuals who belonged to 3783 complete twin pairs with available biological material, 39 twin pairs were discordant for incident AMI. These 78 individuals (39 twin pairs) were included in the analyses. Finally, information regarding dates of birth and death was obtained from the Danish Civil Registration System Registry (see https://cpr.dk for details). Out of the 78 individuals under study, three individuals died during the 3 years of follow-up; all three individuals were twins holding an AMI diagnosis.

Informed consents were obtained from all survey participants, and the survey, as well as the laboratory analyses (see below), were approved by the Regional Committees on Health Research Ethics for Southern Denmark (S-VF-19980072 and S-20200214). The study was conducted in accordance with the Declaration of Helsinki.

### 4.2. Laboratory Analyses

Frozen blood plasma samples were thawed and used for proteome analysis by nano-liquid chromatography coupled to tandem mass spectrometry (nano-LC-MS-MS). The plasma content of the nicotine metabolite cotinine was measured by LC-MS-MS using an in-house method. Plasma levels of total cholesterol, triglycerides, high-density lipoprotein (HDL), and low-density lipoprotein (LDL) were measured using immunochemistry (Roche Cobas 8000, Basel, Switzerland). Prior to statistical analysis (see below), non-HDL was calculated by subtracting HDL levels from the total cholesterol level (as carried out in [43]).

### 4.3. Proteome Analyses

Plasma proteins (5 μg) were reduced (5 mM dithiothreitol, 50 °C, 30 min) and alkylated (15 mM iodoacetamide, 30 min, room temperature in the dark) and digested overnight (0.25 trypsin µg, 37 °C overnight). Enzymatically digested peptides were randomly labeled with either of the TMT reagents 127N, 127C, 128N, 128C, 129N, 129C, 130N, or 130C. A pool of all samples was tagged with reagent 126 and served as an internal standard. Prior to LC-MS-MS analysis, the resulting TMT sets were fractionated by high-pH reversed-phase fractionation with a linear gradient of 25 min from 10% solvent B (20 mM ammonium formate in 80% acetonitrile (ACN), pH 9.3) to 55% solvent B at a 6 µL/min flowrate on a Dionex Ultimate 3000 RSLCnano system inline coupled to a Dionex 3000 Ultimate UV detector (210 nm) and a Dionex Ultimate 3000 autosampler configured as a fraction collector (Thermo Scientific, Bremen, Germany), yielding seven fractions per TMT set to be analyzed with nano-LC-MS-MS.

Nano-LC-MS-MS analysis of the fractionated samples was conducted virtually as previously described [44,45] using an Orbitrap Eclipse mass spectrometer (Thermo Fisher Scientific, San Jose, CA, USA) equipped with a FAIMS interface. Prior to analysis, the peptide fractions were separated using a nano-HPLC interface (Dionex UltiMate 3000 nano-HPLC, Thermo Scientific, Bremen, Germany). Briefly, peptide samples were trapped by a custom-made fused capillary pre-column (2 cm length, 360 µm OD, 75 µm ID packed with ReproSil Pur C18 5 µm resin (Dr. Maish, GmbH, Ammerbuch, Germany)) with a flow of 4 µL/min for 8 min followed by separation on a custom-made fused capillary column (25 cm length, 360 µm OD, 75 µm ID, packed with ReporSil Pur C13 1.9 µm resin (Dr. Maisch, Ammerbuch-Entringen, Germany)) using a linear gradient ranging from 88–86% solution A (0.1% formic acid, Fluka, Seetze, Germany) to 27–32% B (80% acetonitrile (J.T. Baker, Gliwice, Poland) in 0.1% formic acid) over 119 min.

Mass spectra were acquired in positive ion mode by switching between CVs of −50 V and −70 V, applying an automatic data-dependent switch between an Orbitrap survey MS scan in the mass range of 400 to 1200 *m*/*z*, followed by peptide fragmentation applying normalized collisional energy of 40% in a 2-s cycle. MS1 spectra were acquired at a resolution of 60,000 at 200 *m*/*z* by applying a dynamic exclusion of previously selected ions for 60 s. MS2 spectra were acquired at 50,000 resolution at 200 *m*/*z* with a normalized AGC target of 200% using a 0.4 *m*/*z* isolation window. The maximum injection time was 86 ms. The MS1 AGC target was 400,000 ions, and the MS2 AGC target was 100,000 ions. The resulting raw files were processed using Proteome Discoverer software (version 2.4.0.305) virtually as previously described [46].

Protein measurements are given as ratios of the actual amount of protein in a plasma sample to the average amount in a calibrator-plasma pool generated from all plasma samples analyzed. The two co-twins in each twin pair were analyzed in the same MS plex. In total, 869 proteins were measured with nano-LC-MS-MS; after removing proteins with no or off-scale measurement values and cleaning the data [47], 715 proteins were left for analysis.

Finally, for experimental validation of the seven proteins passing correction for multiple testing (i.e., the proteins in Table 2), we performed parallel reaction monitoring mass spectrometry (PRM-MS) of these proteins in 32 individuals for whom nano-LC-MS-MS data were available for a minimum of six of the seven proteins. Moreover, we also analyzed the APOA1 and APOB proteins as positive controls; APOA1 and APOB are often investigated in relation to CVD and are known to be correlated with HDL and LDL values, respectively. APOA1 and APOB had been measured in all individuals by the nano-LC-MS-MS method in the present study. PRM-MS analysis was performed on an Orbitrap Eclipse Tribrid mass spectrometer (Thermo Fisher Scientific). Peptides were trapped and separated on the same column configuration as previously described, using a 36-min linear gradient from 91% Buffer A (0.1% FA in water) to 32% Buffer B (0.1% FA, 80% ACN/water). Precursors were monitored in an unscheduled manner using the quadrupole to isolate precursors with an isolation width of 0.7 *m*/*z*, followed by fragmentation using HCD (NCE = 30%). The measurement of the fragments was performed in the Orbitrap at 15,000 resolution. The generation of the isolation list for the previously identified proteins as well as raw data processing were performed in Skyline (v. 21.2.0.536). The isolation list is available in Appendix A. The nano-LC-MS-MS values and PRM-MS values for the same individuals were plotted in two-way scatter plots, and correlation coefficients were calculated. Finally, with the aim of obtaining a *p*-value for the correspondence between the two MS measurements, a linear regression model was fitted.

### 4.4. Survey Data

In addition to the registry and laboratory data, survey data regarding information on body mass index (BMI), systolic blood pressure, diabetes disease status, medication use, zygosity, and smoking status were included in the present study. BMI (kg/m^2^) was derived from measured height and weight: weight was measured once, while height was measured twice; thus, for height, an average of the two measurements was used. Systolic blood pressure was measured as the standard sitting blood pressure of the upper arm. The blood pressure was measured twice with a one-minute break between, and an average of the two blood pressure measurements was used. The status of current diabetes type 1 or 2 was based on the question, “Which serious, prolonged diseases do you have now or have had previously? If a doctor has ever diagnosed you with one of the diseases mentioned below, please answer “yes”. Medication use of (a) diabetes medication, (b) heart- and blood pressure medication, (c) coagulation inhibitory medication, or (d) lipid lowering medication was based on self-report of all medication taken. Medication was grouped based on the ATC guidelines for 2020 (https://www.whocc.no/, 23rd edition from January 2020). The zygosity of the twin pairs was based on four questions regarding physical similarity, which previously have been shown to correctly classify more than 95% of the pairs as compared to genetic markers [48]. Information concerning smoking status (never/former, or current smoker) was obtained from questionnaire data and verified by the cotinine measurements; out of the 78 individuals, 56 displayed a cotinine value ≤ 1.7 ng/mL, while 22 individuals held ≥ 27 ng/mL. For 75 out of 78 individuals, the cotinine measurement corresponded to the self-reported smoking status; consequently, one individual was transferred from the current smoker group to the never/former smoker group, while two individuals were transferred from the never/former smoker group to the current smoker group. Lastly, for diabetes status and systolic blood pressure, a maximum of 2 out of the 78 individuals had no data. These values were imputed by the most frequent answer (diabetes status) and, respectively, the mean of the study population (systolic blood pressure). One height measurement was corrected from 5.86 m to 1.86 m.

Information on the gender of the study participants was based on information from the Danish Patient Registry; in the registry, the information on gender is based on midwife journals, unless an individual has applied to the Danish Civil Registration System Registry for a legal change of gender. In the present study, we did not perform gender stratified analysis as the sample sizes of such analysis stratified by zygosity would not enable a comprehensive analysis; the 13 dizygotic opposite-gender twin pairs are not applicable for such analysis, while for the 12 monozygotic twin pairs, the 12 dizygotic same-gender twin pairs, and the 2 twin pairs of unknown zygosity, the gender distribution was 6 male pairs (50%) and 6 female pairs (50%); 9 male pairs (75%) and 3 female pairs (25%); and 2 male pairs (100%), respectively.

### 4.5. Preparation and Imputation of Proteome Data for Statistical Analysis

Of the 715 proteins for which measurements were obtained in the nano-LC-MS-MS analysis, 197 proteins had complete data (i.e., had a call rate of 100%), while for the remaining 518 proteins, 63 proteins had a call rate of >75% to <100%, 103 proteins had a call rate of >50% to <75%, 154 proteins had a call rate of >25% to <50%, and 198 proteins had a call rate of >0% to <25%. With the aim of analyzing as many proteins as possible, inverse probability weighting [49] was used prior to Cox regression analysis for imputing values for the proteins not measured in all individuals. The ipw library in R version 4.1.0 was used. Initially, different cut-offs regarding the call rate for the proteins (i.e., the percentage of individuals holding data for a given protein) were evaluated for the imputation method. Given the sample size of the study population and the interest in investigating the intra-pair differences between co-twins with an AMI and co-twins without an AMI in all twins, as well as in the monozygotic twins, only proteins with a call rate of 50% or above were included, as the statistical models appeared most stable with this cut-off. Furthermore, to ensure the stability of the intra-pair analysis, the protein values were multiplied by 100 before analysis (see below). The cut-off of 50% for protein call rate and the multiplication by 100 before analysis were initially determined by thoroughly investigating the protein data grouped by 10% decreasing call rate and multiplication by either 10 or 100, solely inspecting the stability of the calculations in R, and not inspecting the results of the statistical analysis.

### 4.6. Statistical Analyses

The association between the protein values and time to AMI diagnosis was analyzed using Cox regression (using the survival library in R); the study population was analyzed as twin pairs employing a stratified Cox model, where the baseline hazard functions were twin pair specific (strata on twin pair ID). This model investigates the intra-pair differences in protein levels between the twin with AMI and the co-twin without AMI. This means that a hazard ratio (HR) above 1 can be interpreted as an increased level of a given protein in the AMI twins relative to the non-AMI twins, and a HR below 1 can be interpreted as a decreased level of a given protein in the AMI twins relative to the non-AMI twins. To ensure proper age adjustment and allow non-linear relations between the protein levels and the risk of AMI, a Cox regression model with age as a timescale was performed. We included age at blood sampling, which defines age at delayed entry, age at diagnosis/end-of-follow-up as the status age, and AMI status as the status variable. It was not relevant to include age at death as status age in the analysis as only individuals getting an AMI diagnosis died during follow-up. To ensure the stability of the intra-pair analysis, the protein values were multiplied by 100 before analysis. Following the definitions by the European Association of Preventive Cardiology [43], we considered the following co-variates in the analyses: gender, smoking status (current vs. former or never smoker), systolic blood pressure, and non-HDL level. Age at blood sampling is adjusted for by including it as the entry age. This model was used in the analysis of all proteins. The Cox proportional hazard assumption was tested, and the assumption was fulfilled considering the number of proteins tested. Finally, the adjustment for multiple testing was performed by the Benjamini and Hochberg false discovery rate (FDR) correction method [50]. In the study population, 12 twin pairs were monozygotic (MZ) twin pairs, hence enabling the study of intra-pair differences in protein levels in genetically identical individuals. This analysis is robust as it excludes shared confounding, particularly the potential genetic confounding by design. A sub-analysis of these MZ twin pairs and the proteins with a *p* value below 0.05 in all twin pairs was performed with the aim of confirming that the direction of effect was not influenced by genetic variation. All analyses were conducted in R version 4.1.0.

### 4.7. Bioinformatic Analyses

Gene Ontology (GO), STRING cluster, *Kyoto Encyclopedia of Genes and Genomes* (KEGG), and Reactome enrichment analyses were conducted for proteins displaying a *p* value below 0.05 in the association analysis. These bioinformatic analyses were performed using the STRING database for functional protein association networks (https://string-db.org, [51] (accessed on 11 August 2023)). Some of the immunoglobulins in the present dataset were not annotated in the STRING database and were consequently not included in the bioinformatic analyses. With respect to GO analysis, biological processes, molecular functions, and cellular components were inspected. Moreover, hierarchical groups were identified; for KEGG, the Brite hierarchy groups were identified via the KEGG database (https://www.genome.jp/kegg (accessed on 17 August 2023)); for Reactome, the Event hierarchy groups were found (https://reactome.org/content/toc (accessed on 17 August 2023)); and for GO terms, the overall grouping of GO terms was examined by inspection of ancestral charts via the QuickGO database (https://www.ebi.ac.uk/QuickGO (accessed on 15 August 2023)). The GO terms were grouped by common ancestor term at the top of the ancestor chart, with the purpose of reflecting the common biology of the terms identified. The STRING clusters were grouped according to their hierarchy clustering (https://string-db.org/cgi/download.pl (accessed on 11 August 2023)). Finally, protein–protein interaction (PPI) networks were investigated using the STRING database.

## Figures and Tables

**Figure 1 ijms-25-02638-f001:**
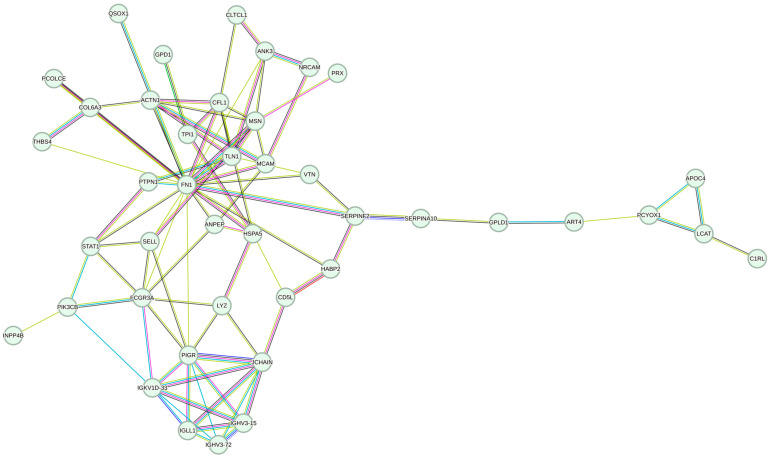
Protein–protein-interaction network of the 57 proteins with a *p* value below 0.05 in stratified Cox regression analysis of all 39 AMI discordant twin pairs. Notes: the colors of the edges reflect the following: (1) known interactions; purple; experimentally determined, and turquoise; from curated databases, (2) predicted interactions; green: gene neighborhood, red: gene fusions, dark blue: gene co-occurrence, and (3) others; light green: textmining, black: co-expression, and light purple: protein homology.

**Table 1 ijms-25-02638-t001:** Descriptives of the study population.

Phenotype	Descriptives
No. individuals (no. twin pairs)	78 (39)
Age at blood sampling (years) (range, mean (SD))	48–79, 66.0 (7.9)
Gender (males (%)/females (%))	47 (60%)/31 (40%)
Zygosity per twin pair (%)	12 monozygotic (31%)
	12 same gender dizygotic (31%)
	13 opposite gender dizygotic (33%)
	2 unknown zygosity (5%)
Time from blood sample to AMI diagnosis (years) (range, mean (SD))	0.01–2.9, 1.4 (0.8)
Smoking status (never/former (%), current (%))	56 (72%), 22 (28%)
Systolic blood pressure (mmHg) (range, mean (SD))	116.5–220.5, 156.2 (22.6)
Lipid levels (mmol/L):	
Total cholesterol (range, mean (SD))	2.9–8.1, 5.5 (1.2)
Triglycerides (range, mean (SD))	0.6–10.8, 2.0 (1.3)
High-density lipoprotein (range, mean (SD))	0.5–2.4, 1.4 (0.2)
Low-density lipoprotein (range, mean (SD))	0.9–5.5, 3.1 (1.0)
Self-reported medication use (%)	
(1) Heart and blood pressure/coagulation inhibitory/lipid-lowering medication	48 (62%)
(2) Diabetes medication	1 (1%)
(3) Non-users	29 (37%)
Body mass index (kg/m^2^) (range, mean (SD))	20.1–39.6, 27.9 (3.7)
Self-reported diabetes disease status (%)	1 type–1 diabetic (1%)
	6 type–2 diabetics (8%)
	71 non-diabetics (91%)

Notes: No: number of; SD: standard deviation; AMI: acute myocardial infarction; kg: kilograms; m: meter; mmHg: millimeter of mercury; mmol: millimole; L: liter.

**Table 2 ijms-25-02638-t002:** The seven proteins passing correction for multiple testing in stratified Cox regression analysis of the 39 AMI discordant twin pairs.

Accession No.	Protein Description	Gene	HR	SE	*p*-Value	95% CI	q-Value	Imputed (Protein Call Rate (%))
Q6ZV73	FYVE, RhoGEF, and PH domain-containing protein 6	*FGD6*	37.26	487.3	7.90 × 10^−224^	(29.84, 46.52)	2.87 × 10^−221^	Yes (64.1)
P43121	Cell surface glycoprotein MUC18	*MCAM*	22.57	473.6	4.05 × 10^−183^	(18.26, 27.89)	7.36 × 10^−181^	Yes (51.3)
P42338	Phosphatidylinositol 4,5-bisphosphate 3-kinase catalytic subunit beta isoform	*PIK3CB*	1.04	0.03	6.70 × 10^−5^	(1.02, 1.06)	8.11 × 10^−3^	Yes (61.5)
P18428	Lipopolysaccharide-binding protein	*LBP*	0.91	0.06	2.27 × 10^−4^	(0.86, 0.95)	0.017	Yes (74.4)
A0A0B4J1V0	Immunoglobulin heavy variable 3-15	*IGHV3-15*	0.96	0.02	2.34 × 10^−4^	(0.94, 0.98)	0.017	Yes (74.4)
Q9NZP8	Complement C1r subcomponent-like protein	*C1RL*	0.93	0.04	5.57 × 10^−4^	(0.90, 0.97)	0.032	Yes (87.2)
P55056	Apolipoprotein C-IV	*APOC4*	0.95	0.03	6.14 × 10^−4^	(0.93, 0.98)	0.032	Yes (87.2)

Notes: APOC4: apolipoprotein C-IV; C1RL: complement C1r subcomponent-like protein; FGD6: FYVE, RhoGEF, and PH domain-containing protein 6; HR: hazard ratio; IGHV3-15: immunoglobulin heavy variable 3-15; LBP: lipopolysaccharide-binding protein; MCAM: cell surface glycoprotein MUC18; PIK3CB: phosphatidylinositol 4,5-bisphosphate 3-kinase catalytic subunit beta isoform; q-value: false discovery rate corrected *p* value; SE: standard error; 95% CI: 95% confidence interval. Call rate: percentage of individuals holding data for a given protein (imputation of missing protein data was performed by inverse probability weighting).

**Table 3 ijms-25-02638-t003:** Reactome and KEGG pathway analyses of the proteins displaying a *p* value below 0.05 in stratified Cox regression analysis of the 39 twin pairs.

Hierarchy Group	Pathway ID	Description of Pathway	Observed Gene Count	Background Gene Count	Strength	FDR	Matching Proteins in the Network
Reactome							
Hemostasis	HSA-109582	Hemostasis	13	607	0.91	4.2 × 10^−6^	TLN1, SERPINF2, HSPA5, IGLL1, FN1, QSOX1, PTPN1, ACTN1, CFL1, JCHAIN, CFD, SELL, PIK3CB
	HSA-114608	Platelet degranulation	8	126	1.38	4.2 × 10^−6^	TLN1, SERPINF2, HSPA5, FN1, QSOX1, ACTN1, CFL1, CFD
	HSA-76002	Platelet activation, signaling, and aggregation	10	260	1.16	4.2 × 10^−6^	TLN1, SERPINF2, HSPA5, FN1, QSOX1, PTPN1, ACTN1, CFL1, CFD, PIK3CB
	HSA-202733	Cell surface interactions at the vascular wall	5	139	1.13	0.0108	IGLL1, FN1, JCHAIN, SELL, PIK3CB
Hemostasis/Signal Transduction	HSA-354192	Integrin signaling	3	27	1.62	0.0169	TLN1, FN1, PTPN1
Signal Transduction	HSA-186797	Signaling by PDGF	4	58	1.42	0.0088	COL6A3, THBS4, STAT1, PIK3CB
Developmental Biology	HSA-9675108	Nervous system development	9	575	0.77	0.0088	ANK3, COL6A3, TLN1, PRX, MSN, NRCAM, CFL1, CLTCL1, PIK3CB
	HSA-422475	Axon guidance	8	551	0.74	0.0187	ANK3, COL6A3, TLN1, MSN, NRCAM, CFL1, CLTCL1, PIK3CB
Immune System	HSA-168256	Immune System	16	1979	0.49	0.0108	LBP, VTN, LYZ, ANPEP, HSPA5, FN1, PIGR, MSN, STAT1, QSOX1, FCGR3A, PTPN1, CFL1, CFD, SELL, PIK3CB
	HSA-168249	Innate Immune System	11	1041	0.6	0.0169	LBP, VTN, LYZ, ANPEP, PIGR, QSOX1, FCGR3A, CFL1, CFD, SELL, PIK3CB
Extracellular matrix organization	HSA-3000170	Syndecan interactions	3	27	1.62	0.0169	VTN, FN1, ACTN1
KEGG							
Cellular Processes; Cellular community—eukaryotes	hsa04510	Focal adhesion	7	195	1.13	3.2 × 10^−4^	VTN, COL6A3, TLN1, THBS4, FN1, ACTN1, PIK3CB
Cellular Processes; Cell motility	hsa04810	Regulation of the actin cytoskeleton	5	209	0.96	0.0196	FN1, MSN, ACTN1, CFL1, PIK3CB
Environmental Information Processing; Signaling molecules and interaction	hsa04512	ECM-receptor interaction	4	88	1.24	0.0169	VTN, COL6A3, THBS4, FN1
Human Diseases; Infectious disease: viral	hsa05165	Human papillomavirus infection	6	324	0.85	0.0196	VTN, COL6A3, THBS4, FN1, STAT1, PIK3CB
Human Diseases; Cancer: overview	hsa05205	Proteoglycans in cancer	5	194	0.99	0.0196	VTN, ANK3, FN1, MSN, PIK3CB

Notes: ACTN1: alpha-actinin-1; ANK3: ankyrin-3; ANPEP: aminopeptidase N; CFD: complement factor D; CFL1: cofilin-1; CLTCL1: clathrin heavy chain 2; COL6A3: collagen alpha-3(VI) chain; ECM: extra cellular matrix; FCGR3A: low-affinity immunoglobulin gamma Fc region receptor III-A; FDR: false discovery rate; FN1: fibronectin; hsa/HSA: *homo sapiens*; HSPA5: endoplasmic reticulum chaperone BiP; IGLL1: immunoglobulin lambda-like polypeptide 1; JCHAIN: immunoglobulin J chain; LBP: lipopolysaccharide-binding protein; LYZ: lysozyme C; MSN: moesin; NRCAM: neuronal cell adhesion molecule; PDGF: platelet-derived growth factor; PIGR: polymeric immunoglobulin receptor; PIK3CB: phosphatidylinositol 4,5-bisphosphate 3-kinase catalytic subunit beta isoform; PRX: periaxin; PTPN1: tyrosine-protein phosphatase non-receptor type 1; QSOX1: sulfhydryl oxidase 1; SELL: L-selectin; SERPINF2: alpha-2-antiplasmin; STAT1: signal transducer and activator of transcription 1-alpha/beta; THBS4: thrombospondin-4; TLN1: talin-1; VTN: vitronectin.

## Data Availability

According to Danish and EU legislation, the transfer and sharing of individual-level data requires prior approval from the Danish Data Protection Agency and requires that data sharing requests be dealt with on a case-by-case basis. Therefore, the data from the present study cannot be deposited in a public database. However, we welcome any inquiries regarding collaboration and individual requests for data sharing.

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
