# Peer review of "A Mass Spectrometry-Based Proteome Study of Twin Pairs Discordant for Incident Acute Myocardial Infarction within Three Years after Blood Sampling Suggests Novel Biomarkers"

_ijms, 2024, doi:10.3390/ijms25052638_

Round 1

Reviewer 1 Report

Comments and Suggestions for Authors

The authors present a comprehensive study including twin pairs discordant for AMI within 3 years. They perform proteomic analysis of blood samples collected previously to the AMI event. In my opinion the study is well designed but more individuals especially from monozygous twin pairs should be included.

As major concerns, a detailed explanation about the design of the Cox analysis should be implemented. i. e. what confounding factors are taking into account in each regression and what results correspond to each factor in the multivariate analyses. No information is given about what variables reach statistical significance in the univariate analyses prior to the multivariate one.

More explanation is needed about the selection of the call rate of 50% for the inclusion of proteins for the statistical model. How is the comparison when using 75%? It doesn’t seem a statistically robust method for the selection.

Since conclusions are mainly focused on monozygous twin pair, perhaps a bigger sample size is needed. As the authors recon, no sex analyses may be implemented because of the low sample size in this subgroup. Since no explanation is given about the three years of AMI discordance, perhaps this period should be modified in order to include more monozygous twin pairs?

Author Response

Response to comments regarding monozygotic twins and changes made:

Thank you very much for your time to review our manuscript and the constructive comments to help us improve our work. Regarding the monozygotic twins, we realize that we need to be clearer in our description of the study population and the aspects of monozygotic twins. We have hence added elaborations on this (see below).

The basis of the present study was a population-based and nationwide epidemiological survey (the INFRA cohort) of 12,359 Danish twins surveyed by the Danish Twin Registry (DTR). The DTR holds basic information on all Danish twins back to 1870, making it the oldest, and among the largest nationwide twin registries in the world. The INFRA cohort conducted from 2008 to 2011 is the largest cohort with blood sample collection performed by the DTR to date. To the best of our knowledge no bigger twin cohort with biological data exists in Denmark. The 39 twin pairs investigated here were identified by linking the 12,359 twins of the cohort to diagnoses from the National Danish Patient Registry (DPR) covering diagnoses on the entire Danish population back to 1977. As our focus in the present study was on the short/medium term AMI cases (as there in general is a lack of studies on such biomarkers) we chose to investigate the cases within the three years, knowing that some of them would be dizygotic. Furthermore, due to Danish and EU legal regulations (the General Data Protection Regulation) at the time of conducting the present study, data from the DPR was available from the start of the DPR until 2014, giving a maximum possible follow-up of seven years after blood sampling. Hence, increasing the time period to the maximum would not have given many more monozygotic twins pairs. Finally, we are very much aware that the sample size of 12 monozygotic twin pairs does not enable stratified analysis or bold conclusion, this is also why we throughout the manuscript (including the abstract) refer to the findings in all twins, and describe monozygotic twins as ‘echoing’ these findings (i.e., the direction of effect in the monozygotic twins in general showed good correspondence to all twins for 47 out of 54 proteins investigated).

We have now elaborated in the Introduction regarding the statistical power of monozygotic twins and the number of monozygotic twins investigated here (page 2, lines 91-96 and page 3, lines 115-121), in the Materials and Methods section regarding the identification of the 39 twin pairs and the rationale for the timeline (page 12, lines 425-445), in the Materials and Methods section regarding the description of gender (page 15, lines 570-575). Finally, we have in the Discussion section elaborated on the aspects of the monozygotic twin pairs (page 11, lines 368-369 + 384-390).

Response to comments regarding Cox regression analysis:

We very much appreciate these comments, as we realize that we need to be clearer in our description of the statistical method. We have now elaborated in the Materials and Methods section (page 15, line 609 to page 16, line 625).

In the present study we have chosen to follow the practice of including the co-variates known to be relevant confounders for a given outcome based in a priori subject-matter knowledge and not from statistical association in the data analyzed (Hernan et al. 2002); here the a priori subject-matter knowledge being the recommendations by the European Association of Preventive Cardiology. Furthermore, if we classically should compare crude estimates to adjusted or stratified estimates in order to thoroughly inspect the potential confounders, we would need to perform (363 proteins + 1 AMI) x 5 co-variates x (crude and adjusted/stratified models) = 3640 tests, which would be difficult to do in a consistent manner. At the same time, we would like to apply one statistical model (i.e., one set of co-variates) in the association analysis in order to be able to compare across proteins. Finally, as the present study perform intra-twin-pair analysis, the intra-pair differences in the potential confounder can potentially be small, e.g., age at blood sampling, and potentially be insignificant if evaluated on their own. That is the reasoning for our choice to include the risk factors put forward by the European Association of Preventive Cardiology as the co-variates in the present study.

(Hernán, Hernández-Díaz, Werler, Mitchell, “Causal Knowledge as a Prerequisite for Confounding Evaluation: An Application to Birth Defects Epidemiology”, American Journal of Epidemiology, Vol. 155, No. 2, 2002)

Response to comments regarding the cut off for imputation:

We very much appreciate the comment, as we realize that we need to be clearer in our description of this cut off. We have now elaborated in the Materials and Methods section (page 15 lines 588-596).

The cut off of 50% was determined by initially grouping the protein data by 10% decreasing call rate (i.e., 100, <100-90, <90-80, <80-70, <70-60, <60-50, <50-40, <40-30, <30-20, <20-10, <10) and  varying the value, which was multiplied to the protein values before statistical analysis. This setup was thoroughly inspected in both all twins, as well in the monozygotic twins. Protein values multiplied by 100 and using a cut of call rate above 50% showed the most stable calculations in R. Finally, we have in the results of the present study included the call rate of the proteins (for all proteins in the Supplementary Results), and as such the reader can sort by call rate and inspect all proteins with a call rate above 50%, including proteins with a call rate above 75%, or whatever call rate might be of interest.

Reviewer 2 Report

Comments and Suggestions for Authors

Dear Authors,

Congratulations on your  interesting and important work regarding biomarkers for AMI. The paper is technically sound and easy to follow, results are appropriately discussed. There are a few aspects that should be considered:

-        Introduction part: there is strong recommendation to move part (lines 97-108) to methodology part.

-        is there any previous similar research regarding biological biomarkers for AMI in Denmark using twin pair desing? Or is this manuscript the first one to discuss this topic? This could be shortly discussed at the end in the Introduction section. 

Author Response

Response to comments:

Thank you very much for your time to review our manuscript and the constructive comments to help us improve our work. We realize that the section at the end of the Introduction is redundant considering the information also given in the Materials and Method section. Consequently, we have deleted the section and added a few details on page 12.  To the best if our knowledge the present study is the first of its kind, not only investigating Danish twins, but also internationally. We have added on page 3, lines 119-121.

Round 2

Reviewer 1 Report

Comments and Suggestions for Authors

All issues have been properly answered.